# What Early User Involvement Could Look Like—Developing Technology Applications for Piano Teaching and Learning

**Tina Bobbe** [1,2,*], **Luca Oppici** [2,3], **Lisa-Marie Lüneburg** [1,2], **Oliver Münzberg** [1,2], **Shu-Chen Li** [2,4], **Susanne Narciss** [2,3], **Karl-Heinz Simon** [5], **Jens Krzywinski** [1,2] **and Evelyn Muschter** [2,4]

1 Industrial Design Engineering, Technische Universität Dresden, 01062 Dresden, Germany; lisa-marie.lueneburg@tu-dresden.de (L.-M.L.); oliver.muenzberg@gmx.de (O.M.); jens.krzywinski@tu-dresden.de (J.K.)
2 Centre for Tactile Internet with Human-in-the-Loop (CeTI), Technische Universität Dresden, 01062 Dresden, Germany; luca.oppici@tu-dresden.de (L.O.); shu-chen.li@tu-dresden.de (S.-C.L.); susanne.narciss@tu-dresden.de (S.N.); evelyn.muschter@tu-dresden.de (E.M.)
3 Psychology of Learning and Instruction, Technische Universität Dresden, 01062 Dresden, Germany
4 Lifespan Developmental Neuroscience, Technische Universität Dresden, 01062 Dresden, Germany
5 Piano Department, Carl Maria von Weber Hochschule für Musik Dresden, 01067 Dresden, Germany; kh.simon@simkh.de
* Correspondence: tina.bobbe@tu-dresden.de

**Abstract:** Numerous technological solutions have been proposed to promote piano learning and teaching, but very few with market success. We are convinced that users' needs should be the starting point for an effective and transdisciplinary development process of piano-related Tactile Internet with Human-in-the-Loop (TaHIL) applications. Thus, we propose to include end users in the initial stage of technology development. We gathered insights from adult piano teachers and students through an online survey and digital interviews. Three potential literature-based solutions have been visualized as scenarios to inspire participants throughout the interviews. Our main findings indicate that potential end users consider posture and body movements, teacher–student communication, and self-practice as crucial aspects of piano education. Further insights resulted in so-called acceptance requirements for each scenario, such as enabling meaningful communication in distance teaching, providing advanced data on a performer's body movement for increased well-being, and improving students' motivation for self-practice, all while allowing or even promoting artistic freedom of expression and having an assisting instead of judging character. By putting the users in the center of the fuzzy front end of technology development, we have gone a step further toward concretizing TaHIL applications that may contribute to the routines of piano teaching and learning.

**Keywords:** tactile internet; piano pedagogy; piano learning; user research; human-centered design; multi-modal feedback; transdisciplinary; user experience; technology acceptance

## 1. Introduction

In recent years, various technological solutions have been researched and developed to improve communication, efficiency, efficacy, and healthy practice in piano learning and teaching [1,2]. Some of those novel systems proved to work technically, but they practically never left the lab, let alone entered the practice routines of piano students and teachers [3,4]. Reasons for such an unsuccessful technology transfer are manifold [5]. A critical one is that purely technology-driven products often fail to address certain aspects of users' needs [6]. Including the end users early on in product development is crucial and will increase the likelihood of a product being accepted and used in practice.

The Tactile Internet with Human-in-the-Loop (TaHIL) approach [7] has the potential to move the field of technology promoting piano teaching and learning forward. On the one hand, Tactile Internet can improve the technical aspects of an application, and on the other hand, the human-in-the-loop approach ensures that an application is tailored to the

needs and capacity of the potential user. Tactile Internet refers to "a network or network of networks for remotely accessing, perceiving, manipulating, or controlling real or virtual objects or processes in perceived real time by humans or machines" (IEEE P1918.1 in [8]). In principle, TaHIL applications could enable piano students (and teachers) to remotely (or in cyberspace) interact with each other as naturally as possible by means of wearable devices equipped with sensors and actuators that are directly integrated in textiles [9]. Such wearable devices, if digitally connected with fast, secure, and reliable communication systems that enable zero-latency human-machine interactions and empowered with machine algorithms, could enable personalized and meaningful feedback or instructions. Importantly, a TaHIL approach to technology development will include an assessment of how piano users consider a technology application to be relevant, effective, and accepted for their practice. This user approach in turn could lead to an increased technology acceptance and adoption in the piano community by teachers as well as students.

In the context of developing scientifically proven applications for the piano that will be accepted and thus adopted by users in the future, we propose the pursuit of a human-centered design approach integrating users already at the initial development phase of TaHIL applications. In this paper, we present an example of how users can (and in our opinion should) be included and how user integration informs the initial stage of technology development.

Integrating the user throughout the different stages of the design process allows developers to properly account for the needs and requirements of the target group. The initial design phase, also referred to as the fuzzy front end—the period between first consideration of an opportunity and ready-for-development-judgement of an idea—is critical for the development of accepted applications [10]. The fuzzy front end phase (Figure 1) is characterized by ambiguity and uncertainty, often involves ad hoc decisions, and follows an ill-defined process [11]. Ideas in this phase are highly adaptable and have a huge impact on the success (thus acceptance) of a new product [12]. Users, at this early stage, can help shape the development direction of a new application.

In a pilot study with a small sample of piano users, we explored how end users can be included in the design of piano-related technology applications. We collected information on how piano teachers and students perceive and accept (or reject) potential scenarios of TaHIL technology in their teaching and learning activities, surveying their needs and suggestions. We visualized three potential TaHIL scenarios based on literature in the fields of music pedagogy and motor learning as visual aids to help and encourage study participants to imagine such TaHIL applications in their piano education and rehearsal routines. While obtaining insights into current piano learning practices and gathering opinions and ideas by pianists, we focused on so-called hedonic (or non-instrumental) qualities of future TaHIL applications that go beyond task- and usability-related aspects of human–product interaction. Finally, for each scenario we extracted acceptance factors, added value, and target group, which informed further development phases. By integrating the user in the fuzzy front end of technology development we have gone a significant step further toward a meaningful TaHIL application that hopefully finds its way into the routines of piano students and teachers.

## 1.1. Human-Centered Design Approach

In order to create new products, services or systems that provide an added value for end users, it is essential to integrate the users' perspective and find out about their goals, feelings, abilities, and practices. Human-centered design (HCD) [13,14] is an approach whereby the perspective of potential users drives decision-making processes during product development. While user involvement is increasingly common in later phases of the development process, where for instance users are invited to test the usability of a product, it is less common to involve users at the early phases. However, this is a critical phase since it is when opportunities are defined and early ideas explored [15]. In the fuzzy front end (Figure 1), the solution space is at its maximum, and it is often unknown whether

the outcome of the process will be a product, a service, or an app. The goal of the fuzzy front end is to determine what (and what not) to design and prototype [15]. Early user integration can emerge through ethnographic research methods, such as observations or interviews (designer moves into the field) or through participatory design approaches in which end users actively co-design possible solutions (users move toward the lab) [16].

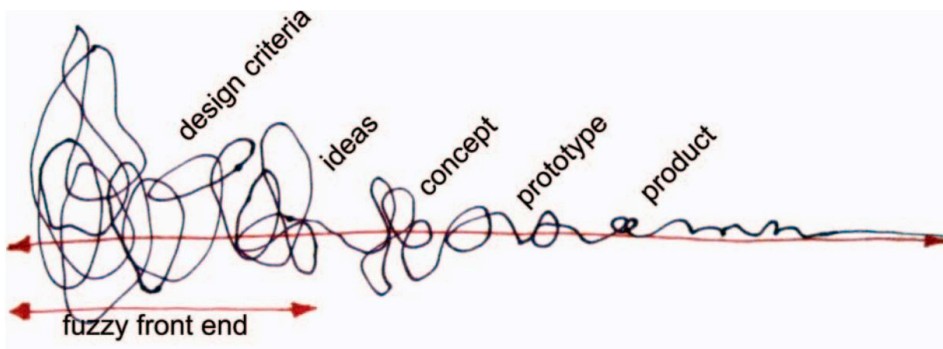

**Figure 1.** The fuzzy front end is characterized by ambiguity and uncertainty [15]. Reprinted with permission of the publisher (Taylor & Francis Ltd., http://www.tandfonline.com).

Complementing HCD, user experience (UX) provides a holistic perspective on human–product interaction [17]. Next to so-called pragmatic (or instrumental) qualities of interaction, which are typically task-related (e.g., effectiveness), are hedonic (or non-instrumental) qualities that go beyond the instrumental value of an interactive system [18]. Hedonic qualities can be categorized in three groups [19]: (1) aesthetic aspects with sub-aspects relating visual, haptic, and acoustic perceptions; (2) symbolic aspects with sub-aspects relating communication and association; and lastly (3) motivational aspects. Future TaHIL gloves, for example, that aim to promote the learning process of piano playing through vibro-tactile feedback in real time could have the following hedonic qualities: wearing the gloves feels good and of high-quality (aesthetic relating to haptic perception), the gloves' appearance represents the piano teachers' identity (communication), the vibro-tactile feedback of the gloves can remind the piano student of their piano teacher by slightly tapping their finger (association), and the gloves motivate the student to practice more often (motivation). Eventually, the perception of both qualities (pragmatic and hedonic) evokes emotions (and vice versa) that result in a user's general opinion of a product as well as future behavior and usage [20]. Hedonic aspects of human–product interaction are particularly relevant in TaHIL technology, whereby applications include wearable devices, close-to-body technology, and embodied feedback. Technologies and humans become more entangled, literally through wearable devices but also regarding the importance of technology in our daily lives [21]. As such, we are convinced that hedonic qualities of human–product interaction play a crucial role in developing human technology. Defining the overall experience and initial values of bodily interactions for end users therefore marks the starting point for TaHIL technology development, as we propose in the following.

*1.2. Technology Acceptance*

While piano playing is a training activity in which emotional expression takes a high value, it is important to understand motivations, hurdles, and fears regarding the use of new technologies such as TaHIL.

Technology acceptance describes an attitude toward a technology and is a core factor influencing a user's behavioral intention to use novel technologies. Technology adoption, however, rather describes the process that goes from the user being aware of technology until making full use of it [22]. We based our investigation on the technology acceptance model (TAM) [23]. The TAM considers perceptions of functionality and usability as core factors related to technology acceptance. The TAM has been applied in a variety of contexts and its factors have been extended to the characteristics of the specific context of interest.

For example, in the context of wearable devices, Wolf et al. [24] showed that in addition to *perceived usefulness* and *perceived ease of use* (both pragmatic qualities), *perceived enjoyment* (relating to hedonic qualities) is another important factor for users' adoption of technology. Aspects, such as *perceived enjoyment* [13,14], *computer playfulness* [15], and other intrinsic motivational factors have been acknowledged by various other studies and contexts as well. While the TAM was not initially designed for exploring the potential willingness of using future technology, it aligns with HCD and UX approaches and can be used in the initial stages of technology development. Next to gathering insights regarding functionality and usability of future TaHIL applications in piano pedagogy, hedonic qualities are nonetheless important to be included since enjoyment, intrinsic motivation, and other emotional aspects might influence users' acceptance of technology.

### 1.3. Proposed User Involvement Process

As aforementioned, hedonic qualities of human–product interaction need to form the starting point for developing TaHIL technologies. We propose the following process, based on the ethnographic approach of user research, for obtaining information on such qualities. First, researchers should familiarize themselves with the context of interest (piano pedagogy) and with the specific problems that end users currently face in their practice. Second, researchers provide early and broad ideas of technological solutions that may tackle some of the end users' routine problems and collect users' opinions and feelings about those solution ideas. Participants are additionally encouraged to propose their own ideas. Information on both steps can be collected via surveys and/or interviews. Lastly, an analysis of users' current problems and user's opinions and ideas regarding proposed solutions will generate requirements for future development of applications. These requirements are based on hedonic qualities and can be used for framing more detailed concepts such as acceptance requirements, added value, and target group to improve acceptance of a future application.

In this study, we created visual scenarios to represent solution ideas for piano-related applications. The scenarios focused on three piano-related domains in which we saw potential in TaHIL technology for tackling them. A review of existing literature and existing applications guided the design of the scenarios (see Section 1.4). These visual aids aimed at encouraging participants to imagine such TaHIL applications in their piano education and rehearsal routine.

### 1.4. Technology Applications for Piano Teaching and Learning

Technological supports are increasingly used in piano practice, and TaHIL technology can provide new opportunities to improve piano teaching, learning, and performance. We envision TaHIL applications to make a significant contribution in at least three critical components of piano teaching and learning: body movement analysis, performance parameter analysis and music visualization. We decided to develop and present participants (potential users) with these visual scenarios rather than giving them existing applications to try for two main reasons: first, TaHIL is a rapidly evolving field and it would be difficult to find an existing application that includes TaHIL's features (i.e., intelligent network of sensors, processors, and actuators); second, we wanted to use scenarios to inspire participants to "look" into the future and not to constrain their view on what already exists.

#### 1.4.1. Body Movement Scenario

The body movement scenario relates to technologies that can capture and analyze finger and (upper-) body movements with high temporal and spatial precision. Coordinating the upper limbs at the required tempo is at the core of playing the piano [25]. An in-depth analysis of how a pianist coordinates the various limbs' degrees of freedom can provide critical information for improving performance and reducing the risk of injury. It is known that a high amount and intensity of rehearsing can lead to overuse syndromes or long-lasting injuries, also referred to as playing-related musculoskeletal disorders [26].

Paying attention to a pianist's movement patterns and teaching more effective techniques to generate energy for the keystrokes can help to reduce the risk of injury [27,28]. In the past, many attempts have been made to precisely capture and analyze body movements during piano playing, either with optical-based systems such as cameras or with other sensor-based systems [27,29]. Sensor-based systems provide the advantage of limiting the problem of occlusion, which happens fairly often due to complex and well-codified fingering patterns [30] such as the thumb-under movement. The use of wearable smart sensors has been proposed. Figure 2 visualizes the body movement scenario with such technology. Here, the pianist wears the smart sensors directly (e.g., connected shirt and gloves). Data is then processed in real time and information on current, past, and suggested movement patterns are available for the user for consultation and feedback. Students and teachers can then use this information to correct movement, improve movement efficiency, and reduce the risk of injury.

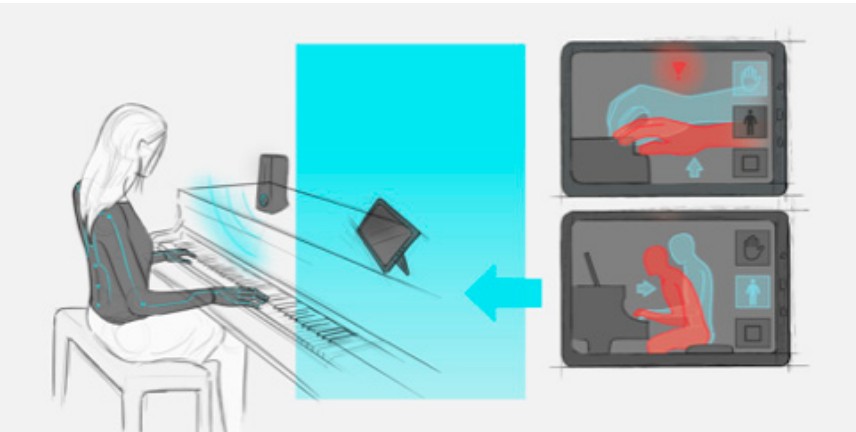

**Figure 2.** Body movement scenario.

### 1.4.2. Performance Analysis Scenario

The performance analysis scenario relates to the analysis of piano performance parameters including accuracy of note playing, consistency in pressure applied to the keys, and consistency in timing [31]. The input data of all available parameters from the piano is analyzed in (quasi) real time using machine learning and in turn can provide feedback on performance. Such feedback can be superimposed on digital musical sheets and/or graphical representations of cord labels (see [32,33]). The processed data is visualized in an app in (quasi) real time for students and teachers and saved for later replay opportunities, providing an additional visualized modality to analyze the just-performed musical piece or excerpt of musical piece. Currently, MIDI-based instruments are capable of providing some information on performance with a detailed visual output as feedback [34,35]. The critical part, currently scrutinized in research, for an accurate assessment of all relevant performance parameters and a suitable outcome is the use of algorithms that are well-trained on annotated corpora (see [33]). Figure 3 depicts what visualized feedback in the performance analysis scenario may look like. This tool could foster how the teacher and the student communicate, make an adjustment to piano playing, and could also assist teachers in understanding what aspects of their students' performance can be improved.

### 1.4.3. Music Visualization Scenario

The music visualization scenario relates to real-time visualization of music or music parts (e.g., melody, rhythm, harmony) that can be created based on sensor, MIDI, or audio data. Such visualizations can be used for artistic purposes or serve to provide feedback on selected quality aspects in music training [36]. In the last few years, attempts have been made to develop applications that are capable of real-time mapping and visualization of music [37,38]. These concepts visually track selected aspects (e.g., low level audio or

musical features such as note onset times or the frequency spectrum) of producing music (singing or instrument playing). TaHIL applications aim at visualizing not only selected aspects of the auditory output but also creating a variety of art forms choreographed to the music, as can be seen in Figure 4. Inspired by research findings indicating that an external focus of attention can be beneficial for motor performance [39,40] we assume that such artistic visualizations may be used to assist pianists in directing their attention externally to the visual output in addition to the auditory output of their musical performance. This might be particularly important when preparing concert performances or auditions.

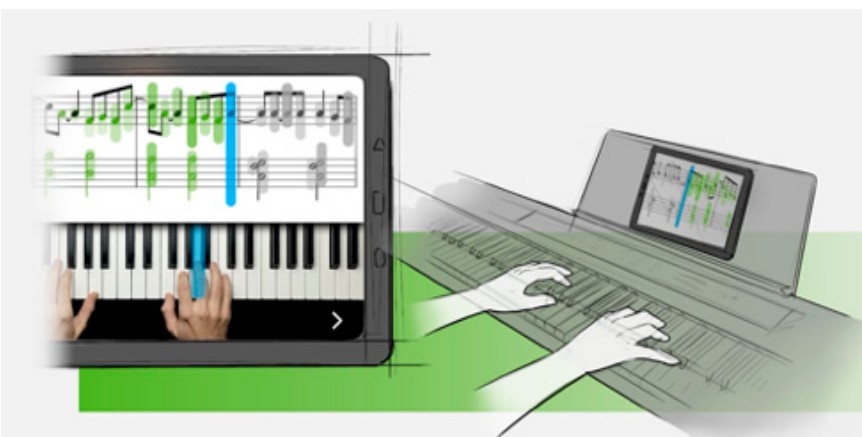

**Figure 3.** Performance analysis scenario.

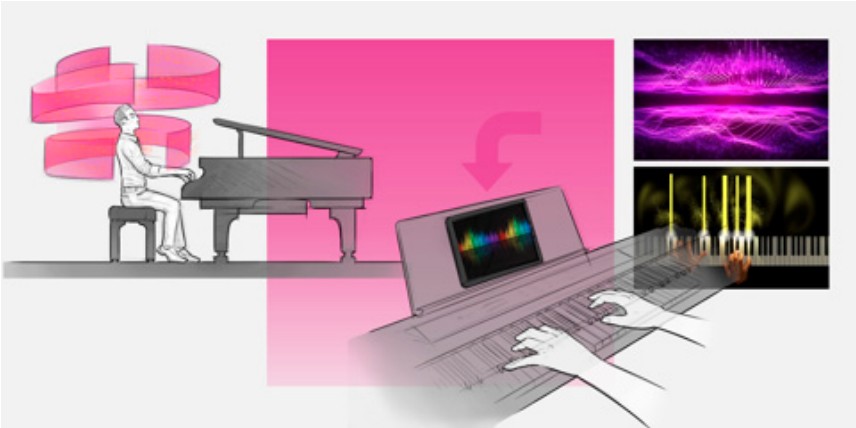

**Figure 4.** Music visualization scenario.

## 2. Evidence for Opinion: User Study Design

We conducted a user study. Since we aimed at obtaining insights into current piano learning practice and gathering opinions and ideas by pianists, we chose the ethnographic approach and conducted interviews (complementing field observations were planned, but could not be carried out due to COVID-19-related restrictions). Through the investigation of piano student's and teacher's needs and opinions regarding future TaHIL applications, we aimed to align our technological research with actual human needs. Our goal was to incorporate a holistic perspective on human–machine interaction, reaching for a technological solution that people are excited about and motivated to use.

### 2.1. Research Aim

The main aim of this study was to follow the proposed process of integrating the user at the initial stage of technology development. This aim can be divided in three sub-aims:

1.  Define current key problems in piano teaching and learning processes

2.  Gather participants' opinions regarding the proposed scenarios
3.  Generate requirements for a future TaHIL application in order to be seen as highly acceptable by piano teachers and students

We focused on hedonic aspects, which may contribute to motivate piano teachers and learners to apply TaHIL technology into their piano learning and teaching process.

### 2.2. Materials and Methods

### 2.2.1. Participants

Participants were selected by purposive sampling. An email was sent to music schools and conservatories in the greater Dresden area (Germany) to invite piano teachers and students to take part in our research. Those who expressed interest were then informed about the details of the research, the online survey, and the interviews. Piano students and piano teachers with different expertise levels were recruited to gain different perspectives on TaHIL applications. Participants provided written informed consent. The study was approved by the Ethical Committee of the Technische Universität Dresden (SR-EK-19012021) and was conducted in accordance with the Declaration of Helsinki.

Twenty-nine individuals participated in the online survey; however, 10 of them did not complete the questionnaire and thus were excluded. As a result, the final sample included 19 participants: 11 teachers and 8 students.

The 11 piano teachers (mean age 42.7 years, 15.8 standard deviation) had on average 15 years of teaching experience (min 0.5 years, max 30 years; 10.7 standard deviation) and had played piano themselves on average for 32 years (min 8 years, max 55 years; 17.4 standard deviation). Five of the teacher respondents considered themselves advanced, whereas 6 considered themselves professionals.

The 8 piano students (mean age 33.9 years, 16.6 standard deviation) had played piano on average for 19.13 years (min 2 years, max 45 years; 14.9 standard deviation). While they practiced piano at home for 2–28 h/week (mean 12.8 h/week, 10.2 standard deviation), a regular piano lesson per week ranged from 0 to 1.5 h (mean 0.9 h/week; standard deviation 0.7). Two of the student respondents considered themselves advanced, whereas 3 considered themselves professionals.

Five volunteers (mean age 32.4 years, 16.3 standard deviation) participated in the semi-structured interviews. Table 1 contains the demographic information of the five interviewees. The three piano teachers had on average 6 years of teaching experience and had played piano themselves on average for 17.3 years. The two piano students had played piano for an average of 7 years.

**Table 1.** Interview participants. Demographic information of interview participants and the respective averages.

| Participant | Status | Gender | Age | Piano Experience in Years | Level | Piano Teaching in Years |
|---|---|---|---|---|---|---|
| Adv1 | Student | Female | 24 | 4 | Advanced | 2212 |
| Adv2 | Teacher | Female | 23 | 19 | Advanced | 1 |
| Beg3 | Student | Female | 61 | 10 | Beginner | 2212 |
| Adv4 | Teacher | Male | 31 | 25 | Advanced | 14 |
| Adv5 | Teacher | Female | 23 | 8 | Advanced | 3 |
| Teacher: mean (standard deviation) | | | 25.7 (4.6) | 17.3 (8.6) | | 6 (7) |
| Student: mean (standard deviation) | | | 42.5 (26.2) | 7 (4.2) | | - |

### 2.2.2. Measures and Data Collection

A mixed-method approach was employed: A quantitative online survey was used to gain insights in piano learning experiences and semi-structured remote interviews were used to gain a deeper understanding of the topic at hand.

Online Survey: The online survey consisted of 15 multiple-choice and nine short-answer questions and contained aspects in three categories. The first question set (14) focused on demographic information of the study population, general learning methods of piano playing such as important performance parameters, training methods, and feedback. The second set contained questions (6) about clinical manifestations and physical as well as psychological complaints related to piano playing. It also included questions about the perception and relation to technical devices in general and for piano-related usage. The final set of questions (4) focused on piano practice during COVID-19 restrictions. For most questions participants were asked to select the best answers offered or record, in free text, their response and/or provide more details to their response (e.g., "Which kind of feedback do you prefer during your piano lesson?": "Interrupt and point out error directly"/"feedback right after the particular played piece"/"feedback at the very end of the lesson"). A translated copy of the survey is provided in Appendix A.

The online survey was distributed to the piano community in January 2021 via email using the online survey software Lime Survey of the Technische Universität Dresden (https://bildungsportal.sachsen.de/umfragen/limesurvey/). The online survey was live for a five-week period.

Semi-structured interviews: Based on the interview questionnaire, a guide with four sets of open-ended questions was developed, following a previously published procedure [41]. Additionally, it provided an introduction of the three scenarios to provoke opinions or inspire participants to come up with ideations of even different TaHIL applications. All questions were open-ended and allowed for the sharing of personal insights and stories beyond the prepared questions. The interviews took place via secured video calls from December 2020 to February 2021. All interviews were audio recorded and transcribed. All data were collected by the same interviewer, held and analyzed in accordance with the Data Protection Act, with the actual data available only to the authors of the present paper.

### 2.2.3. Data Analyses

The online survey data were exported into SPSS V.27 (IBM Corp, Endicott, NY, USA, 2020) and descriptive statistics were calculated for all responses.

The recorded interviews were transcribed and coded in MAXQDA (VERBI Software, Berlin, Germany, 2020). The five interviews resulted in a total of 7.52 h of audio data—an average of 94.21 minutes per interview (the shortest lasted 68 min and the longest 145 min). Thematic analysis was used to analyze the data. Two researchers independently analyzed the interviews and manually coded the text. We started coding deductively with an initial code set based on design and technology acceptance research (e.g., "pragmatic aspects"), the proposed scenarios (e.g., "body movement scenario"), and sentiment to evaluate the nature of a code segment (e.g., "positive impression"). The two researchers coded one interview and inductively added a code group regarding the status quo of teaching and learning (e.g., "physical and psychological stress"). The final code system (see Appendix B) was discussed until consensus was reached. The two researchers continued coding all interviews and discussed the coding segments on 5 occasions until consensus was reached.

In order to answer the first research question ("What are the current key problems in piano teaching and learning processes?"), we started to filter all data against the code "learning/teaching problems", which we used on 89 text segments. We further checked these identified segments against other corresponding sub-codes relating to specific piano playing skills and other categories that emerged during the iterative inductive coding process (e.g., self-directed learning). We ranked the resulting corresponding codes according to the frequency with which they appeared in the interview transcripts. The most frequent occurrences in alignment with "learning/teaching problems" were the sub-codes around

"body posture and movements" (16), "playing by the notes" (13), "online lessons" (12), and "self-directed learning" (8).

To answer the second research question ("What are requirements for a future TaHIL application in order to be seen as highly acceptable by piano teachers and students?"), we filtered the data against each of the scenario codes (e.g., "body movement analysis"). We then checked code relations with the two code groups Technology Acceptance (e.g., "hedonic aspects") and Sentiment (e.g., "positive perception"). We quantified the resulting code relations for an overall assessment (see Figure 4) and summarized the codes for each scenario to gain a thorough understanding.

## 3. Results and Discussion

All questions, responses and quotes were translated from German and distracting filler words were removed for the purpose of this paper (e.g., *"(So) the most common difficulties are probably fingerings. (That is, that) often the fingers just don't do what the children want."*). We will use the nomenclature of the interview participants (e.g., Adv1) from Table 1.

The results showed that (i) most piano performers (students and teachers) experience some sort of physical and psychological issues throughout their practice (Table A1 in Appendix C), (ii) hand and body posture and movements are critical components to consider in piano education, (iii) self-practice at home is problematic for students (lack of proper methods, time, and motivation) (Table 2), and (iv) teachers face many challenges in remote teaching (which is especially relevant in times of COVID-19-related restrictions). With regards to the proposed scenarios, (i) participants expressed general interest in the scenarios, as long as such supports align with pianists' artistic freedom of expression, (ii) they primarily favored haptic feedback over other modalities, and (iii) they were excited about technology that promotes creativity and motivation and had an assisting instead of judging character. This user study led to a more defined solution space for each scenario. The following sections summarize the results, discussions and implications for each scenario, defining (1) current problems, (2) user opinions, (3) acceptance requirement, (4) added value, (5) target group and (6) ideas for future development.

**Table 2.** Reported problems during piano lessons and during independent home practice. Teachers' and students' responses presented as percentages. Multiple responses were possible.

|  | Teachers on Site | Teachers at Home | Students on Site | Students at Home |
|---|---|---|---|---|
| Finding one's way around the keyboard (%) | 27 | 18 | 13 | 13 |
| Agility/finger dexterity (%) | 36 | 36 | 13 | 13 |
| Independence of hands (%) | 55 | 27 | 13 | 38 |
| Playing by notes (%) | 27 | 36 | 13 | 13 |
| Playing by heart (%) | 9 | 9 | 38 | 25 |
| Music theory knowledge (%) | 55 | 9 | 25 | 25 |
| Dynamics/volume (%) | 27 | 9 | 25 | 0 |
| Body posture (%) | 55 | 0 | 0 | 25 |
| Motivation (%) | 18 | 36 | 0 | 38 |
| Lack of time (%) | N/A | 64 | N/A | 63 |

*3.1. Resulting Users' Perspectives and Emerging Acceptance Requirements per Scenario*

3.1.1. Body Movement Scenario

Current problem: The online survey and interviews indicated that almost all teachers and students have experienced piano-playing-related physical stress or overstrain, and that body posture and movements are one of the most urgent learning and teaching problems in current practice, consistent with previous research [26–28].

Users' opinion: Participants did not think that the proposed idea could be beneficial for correcting movements because one ideal technique that fits all individuals does not exist. Pianists' posture and movements are highly individual and can vary between musical pieces or day-to-day and movements are part of the emotional expressions of a musical piece.

> *"Everyone has their own posture at the piano, their own movement in which they feel free, so to speak. That's why you can't say it so precisely: 'Ah here, that's how it is with everyone, that's how you feel good.' Because it's different for everyone." (Adv2)*

When discussing potential feedback modalities for correcting movement, all participants imagined haptic feedback to be more suitable than visual feedback due to the visual focus on notes and piano keys.

Acceptance requirements: The TaHIL application (smart shirt and interface) must address and even promote individuality and artistic freedom of expression regarding movements, posture, and bodies. Processing and feedback characteristics (visual and haptic) must be transparent and easily adjustable by the student and/or teacher.

Added value: The TaHIL application must clearly communicate the overall aim of the pianist's bodily well-being while not hindering creativity and expression. It should further focus on upper-body and arm movements, and thus function without potentially hindering gloves.

Target group: Experts.

Ideas for future development: This scenario could be further developed with a focus on smart machine learning algorithms, which can assess individual body movements and posture (for current advancement in this direction see [42,43]), in order to solve the pianists' concerns. An interesting idea inspired by the interviews was the assessment of the degree of relaxation in specific muscles (e.g., shoulders) to enhance pianists' well-being. It was also highlighted that feedback timing and characteristics should be tailored to the level of the performer's expertise and type of mistake [44]. Vibro-tactile feedback given through a wearable must be well-designed in order to be meaningful and comfortable for the pianist. We assess this scenario as most valuable for professional pianists with high amounts of practice, which can increase the risk of playing-related injuries.

### 3.1.2. Performance Analysis Scenario

Current problems: The online survey and interviews pointed to the most common teaching and learning problems for beginners: finding one's way on the keyboard, independence of hands, reading music, and playing by heart. Additionally, results from the survey reported that nearly half of students have problems at home regarding motivation and suffer from a lack of time for self-practice. This has been supported by interviews and free-text responses from teachers complaining about students not practicing efficiently, with wrong or no methods, or simply not enough.

Users' opinion: Interviewees had divided opinions about this scenario. Similar to the first scenario, four interviewees stressed the importance of room for interpretation within a musical piece. Teachers even encourage their students to bend and break the rules regarding tempo and articulation.

> *"I think that it is also important for students and beginners to learn that the piece doesn't sound the same every day and that it doesn't have to sound the same, and that you can play the same piece 10 times slower than before if you're not feeling well and it will still be music and it can still sound beautiful." (Adv5)*

Acceptance requirements: Visual feedback on performance parameters must go beyond pure note tracking and analyze the pianist's performance and describe (but not judge) it in a visually appealing way in order for pianists to compare and reflect. The application could even encourage the student to interpret pieces differently (e. g., "What story do you want to tell today?"). As a holistic self-practice tool, it can complement piano lessons, increase student's motivation for self-practice, and improve teacher–student-communication.



Added value: The TaHIL concept must clearly communicate the added value for teachers: having the app accompanying piano lessons instead of replacing them and making piano learning a more enjoyable experience.

Target group: Beginners and intermediates.

Ideas for future development: This scenario could be developed further as a self-practice application and therefore act as a holistic learning tool that students use to store their notes, practice assignments, focus on specific skills, record and replay their pieces, and have them analyzed [45]. The teacher could also connect to the app in order to communicate homework assignments (e.g., marking sections directly in the respective note sheet of a recording or record audio/video snippets as comments at specific passages). Furthermore, they and the students themselves would be given the ability to systematically document and evaluate the students' progress. Since motivation appears to be a key problem for piano students' practice routines, this application could incorporate gamification elements [18], depending on age and skill level, to increase their intrinsic motivation to practice.

### 3.1.3. Music Visualization Scenario

Current problems: Most teachers and students reported having experienced psychological problems related to piano playing, primarily stage fright. Given the prominent issue of such performance anxiety, it would be beneficial to develop a TaHIL application that aids pianists in managing or overcoming this fear.

Users' opinion: Participants were skeptical whether this tool could be beneficial regarding their performance on stage. However, they were still very excited about the idea of mapping their sounds into visuals, which could have a highly motivational effect.

> *"But in itself, I think this is a super interesting concept. I don't know how useful it could be as a concentration aid. I think it could be a motivational aid. But then under the aspect that it simply becomes more beautiful the more interesting or multi-layered a work is played. I think that could be an approach, that you simply practice in more detail." (Beg3)*

Acceptance requirements: Visual feedback must go beyond the status quo of musical representation and needs to be highly sensitive, varying in aesthetics, and detect and visualize learning processes.

Added value: The TaHIL concept should focus on the creativity and motivation promoting aspect of this learning and performing aid.

Target group: Intermediates.

Ideas for future development: When developing this concept further we should not exclusively think of it as a tool to improve stage performance but also as a practice tool to increase motivation to play, fine-tune, and interpret. Here, again, underlying AI algorithms are a key element to provide very personalized output that adapts to specific expertise levels and uses gamification elements to remain motivating throughout several stages of learning a new piece of music [18]. With this scenario, we believe that intermediates are the most promising to address since it might be too distracting for beginners and experts might prefer to focus on their inner visualizations.

### 3.1.4. Distance Teaching and TaHIL Gloves Scenario

The distance teaching and TaHIL gloves scenario developed dynamically during the interviews and was added as a fourth scenario.

Current problems: Current pandemic-related circumstances led to novel learning and teaching situations for musicians all over the world. Some of the participants paused their lessons, but most of them continued with the help of video calls or video recordings. All interviewees shared the opinion that distance teaching is of worse quality than face-to-face piano lessons, but better than having no lessons at all. For COVID-19-related reasons or other difficult conditions with limited access (e.g., rural or remote areas, disabilities), improving student–teacher communication in e-learning in musical education is valuable for society.

Users' opinion: This scenario was not shown to the interviewees as a prompt, instead, it emerged when talking about online lessons. TaHIL gloves, worn by a student and teacher, could receive and send tactile and haptic information during online lessons. Three interviewees had a positive impression of this scenario and were excited about this idea, although to two of them it seemed to be almost unimaginable futuristic technology.

*"No, it's something completely different to play with gloves. I think I couldn't play with gloves. And I don't think I would want to learn to play the piano with gloves either."* *(Adv5)*

Acceptance requirements: Gloves must be non-intrusive giving well-designed meaningful vibro-tactile feedback. How the gloves feel and fit and how well the vibro-tactile feedback is designed is most crucial for acceptance. Glove alternatives for the student need to be considered, such as a visually augmented keyboard (AR glasses or key projections).

Added value: This scenario has the most value for beginners, who depend on rudimentary demonstrations, especially during conditions such as those of a pandemic. However, even without considering the impact COVID-19 has had on music learning, this TaHIL application can be highly relevant for democratizing access to piano playing skills and experiences, regardless of geographical location or other limitations [7].

Target group: Beginners.

Ideas for future development: One interviewee had very specific ideas about how to develop and apply TaHIL gloves to remote learning with beginners: to communicate finger movements, orientation on the keyboard, or act as a haptic metronome (for an example see [46]). This serves as a great start for further development of a concept, which can be enriched by gesture recognition features [41].

*3.2. User Research Methods*

Piano playing is a highly emotional, self-identifying form of art. Interviewees' motivation to play the piano goes beyond producing beautiful music; instead, piano playing is their language, word-view, or self-expression. Learning to play the piano was described as a process of self-development. Figure 5 illustrates that interviewees' comments regarding the proposed scenarios happened mostly on a pragmatic level: most aspects had been functionality related (98), followed by aspects related to usability (32). Usability-related aspects were mentioned more often in scenarios of *body movement* and *distance teaching and TaHIL gloves*, where the gloves were part of the scenario. Hedonic aspects were raised the least often (25). When looking at the sentiment metrics of the scenarios, positive and negative aspects were quite balanced with slightly more positive assessments in general (60 positive and 51 negative assessments). The scenario *music visualization* was an exception, where far more positive than negative comments appeared (16 positive and 4 negative assessments). This was also the scenario with the most hedonic aspects (9).

The relatively low number of hedonic responses to scenarios raises the question of whether those ethnographic research methods (such as interviews) were suitable for gathering ideas and requirements on a hedonic level. Nonetheless, participants' perception of the scenario *music visualization* met our goal to excite and intrinsically motivate them to apply TaHIL technology in their learning and teaching practice. Here, the most hedonic comments appeared, and it also had the highest ratio of positive to negative comments. What differentiates the scenario *music visualization* from all others is the low level of intrusiveness (assisting and inspiring in contrast to judging). Furthermore, the visual presentation of this scenario differs to the others, illustrating a holistic (and hedonic) experience (immersion in music), rather than pragmatic goal achievements (e.g., taking the right posture). These effects should be considered in future studies and are something worth researching. In general, however, it must be noted that hedonic aspects are often based on implicit knowledge, which is difficult to verbalize in interviews. Co-design approaches are worthy to be considered for future user involvement studies in early phases of technology research and development. Here, cultural probes [47] and generative techniques [48] make implicit knowledge explicit by eliciting emotional responses from the

participants [49]. However, one must admit that co-design approaches are less scientifically valid and accepted and that ethnographic research methods (like in this paper) are a way to combine scientific and design methods.

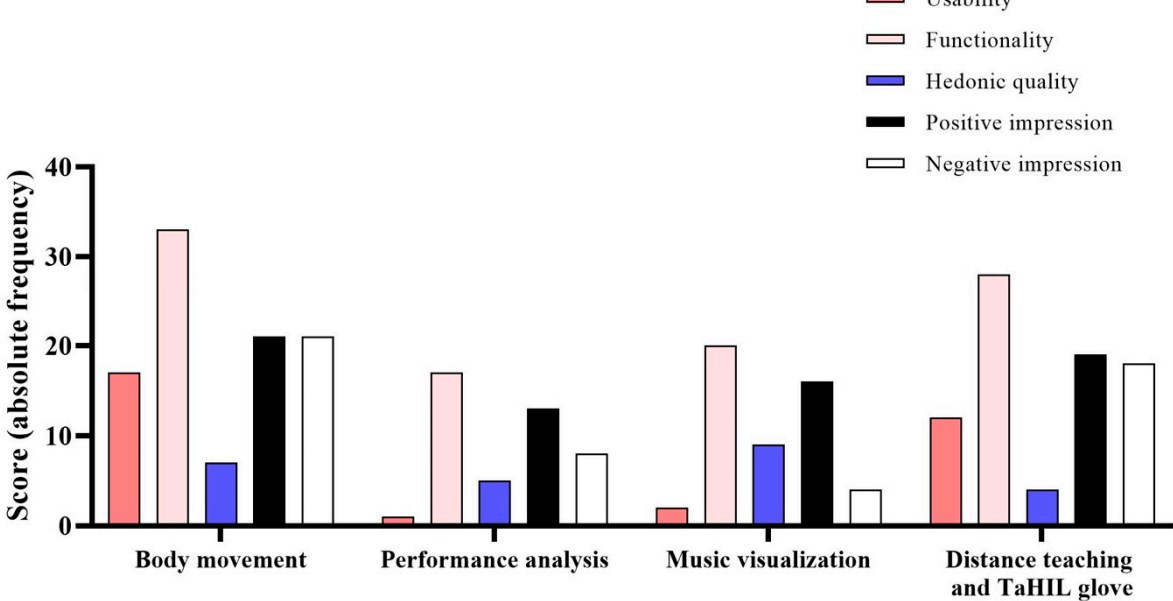

**Figure 5.** Hedonic, pragmatic, and sentiment metrics for the three scenarios. Usability and functionality scores describe pragmatic metrics. The sentiment scores show if the scenario had been discussed in a negative or positive way (positive and negative impression). The scenarios *body movement, performance analysis,* and *music visualization* were described in this paper and used as prompts during the interviews. The scenario *distance teaching and TaHIL gloves* developed dynamically during the interviews and was added to the analysis.

## 4. Summary and Conclusions

In this position paper, we argue for including users in the early stage of the development process of technology applications. Our suggested approach involved collecting insights from potential end users (piano teachers and students) through an online survey and in-depth interviews. We visualized potential literature-based solutions, which served as inspiration for the participants throughout the interviews. This approach resulted in specific acceptance requirements and ideas for future development of piano-related technology applications.

However, results should be considered with care given that data was collected from a small sample of users located in Germany and is not representative of the complete piano community. We also noted that although TaHIL applications have been described as complementary to the current approach and practice of musical education (rather than a replacement to the actual human instructor), acceptance by the general public can be a polemical issue. The most notable fears seem to be that the machine will replace the human, and the loss of artistic beauty. In the present study, some of our participants reported such critical thoughts. It should be noted that our aim was to foster the student–teacher interaction by relieving the teacher's task load, enabling effective communication in e-learning scenarios, and improving students' self-practice and motivation. Furthermore, the early involvement of users and the steady communication of TaHIL science and technology to the public was aimed at answering these concerns.

Throughout this paper we exemplified how interviewing users can help identify their problems and needs, which can refine and direct early ideas toward more meaningful and defined TaHIL applications. This seemingly time-consuming approach of a user study at the fuzzy front end can prevent dispendious work on hard- and software prototypes that do not reflect users' needs and will hardly see light outside of a lab. For example,

participants were very excited about the focus of attention scenario, but rather because they valued the approach of being inspiring and motivating. Another example stemmed from the distance teaching and TaHIL gloves scenario. Some participants stated that they would never wear a glove that even remotely interferes with their finger movements. In this early stage of development, such a harsh critique is bearable since time has not yet been invested in research and development of specific piano playing gloves. Instead, we will focus on alternative ways of augmenting student–teacher interaction, such as using AR glasses.

With regard to the development of piano-related applications, the results can serve as a basis for more extensive and standardized studies. This user study allowed us to obtain an understanding of the current challenges and potentials of each scenario, which will inform the next design steps in developing such TaHIL applications. This work will involve the further development of scenarios by defining concepts, design and development of the application, as well as early prototyping and evaluations with users through protocolized evaluation methods. Here, three perspectives will jointly be taken into account in order to gain a holistic human perspective in technology development [50]: how piano students and teachers perceive these TaHIL prototypes regarding technology acceptance, how the learning process is affected by the TaHIL prototypes, and finally how teachers and students perceive multimodal information from a cognitive perspective. As already mentioned, we aim to continue exploring technology acceptance with further user studies. By actually experiencing a new technology through a prototype, thus involving perceptual motor, cognitive, and emotional elements, participants might form more meaningful intrinsic opinions and effective evaluations of designed concepts. Here, high fidelity of prototypes (close resemblance to a finished product) should be taken into account in order to evaluate hedonic qualities of human–computer interaction, such as aesthetical perceptions [51].

Finally, while our present work focused on potential TaHIL applications for pianists, in principle our approach can be used in a variety of application areas. Although certain aspects are instrument-specific, the idea of including end users at the early stage in order to define key requirements should be applied in a general context of technology development. In such cases, future challenges will include adapting our method to the specifics of the context and setting at hand.

To conclude, we believe that utilizing a holistic approach in the development of new TaHIL applications is crucial. This approach, on the one hand includes transdisciplinary teams of researchers ranging from engineering and informatics to music pedagogy and psychology, while on the other hand keeps the user in the loop. Our main contribution was the inclusion of end users at the center of the fuzzy front end of technology development. With such an approach, we transferred their needs, ideas, concerns, and opinions into specific acceptance requirements to guide further developments of piano-related TaHIL applications. We believe that this approach represents a significant step toward a technology application that will hopefully find its way into the routine of piano students and teachers.

**Author Contributions:** Conceptualization, T.B., E.M. and O.M.; methodology, T.B., E.M. and O.M.; formal analysis, T.B., L.-M.L. and E.M.; investigation, O.M.; data curation, T.B., L.-M.L., E.M. and O.M.; writing—original draft preparation, T.B., E.M. and L.O.; writing—review and editing, T.B., E.M., L.O., L.-M.L., S.-C.L., S.N., J.K. and K.-H.S.; visualization, L.O. and O.M.; supervision, E.M., T.B. and J.K.; funding acquisition, S.-C.L., S.N. and J.K. All authors have read and agreed to the published version of the manuscript.

**Funding:** Funded by the German Research Foundation (DFG, Deutsche Forschungsgemeinschaft) as part of Germany's Excellence Strategy—EXC 2050/1—Project ID 390696704—Cluster of Excellence "Centre for Tactile Internet with Human-in-the-Loop" (CeTI) of Dresden University of Technology.

**Institutional Review Board Statement:** The study was conducted according to the guidelines of the Declaration of Helsinki and approved by the Ethics Committee of Technische Universität Dresden (SR-EK-19012021).

**Informed Consent Statement:** Informed consent was obtained from all subjects involved in the study.

**Data Availability Statement:** Preprocessed data are available upon reasonable request. Data sharing: The authors support data sharing and queries in this regard can be addressed to the corresponding author.

**Acknowledgments:** We thank Prof. Dr. med Hans-Christian Jabusch for valuable discussions and his comments on the developed scenarios. We are grateful to Riccardo Enrico Frink and Anika Fitzer for their support with data collection and preprocessing.

**Conflicts of Interest:** The authors declare no conflict of interest. The funders had no role in the design of the study; in the collection, analyses, or interpretation of data; in the writing of the manuscript, or in the decision to publish the results.

**Appendix A**

Questions and answer options (in italics) for the online questionnaire. If "other" was chosen, participants were instructed to provide details in free text form. All materials were translated from German.

1. Please enter your age in years.
2. For how many years have you been playing piano?
3. How would you describe your skill level on the piano? *Beginner/intermediate/expert*
4. What is your relation to piano playing? *Teacher/student/other*
5. How many hours of piano lessons do you have on average per week? (students only)
6. For how many years have you been teaching piano? (teachers only)
7. How many hours per week do you practice piano at home? (students only)
8. What are your/your students' most common difficulties during piano lessons? *Finding the way around the keyboard/finger dexterity/independence of hands/playing by note/playing by heart/music theory knowledge/dynamics and volume/body posture/motivation/ none/other*
9. What are your/your students' most common difficulties during practice at home? *Finding the way around the keyboard/finger dexterity/independence of hands/playing by note/playing by heart/music theory knowledge/dynamics and volume/body posture/motivation/ lack of time/focus and setting (surroundings)/none/other*
10. What form of practice do you prefer? (students only) *Exploratory free practice/clear instructions and quick feedback*
11. What type of instructor feedback do you/your students prefer during lessons? *Interrupt and point out errors immediately as they occur/feedback right after each musical piece/feedback only at the end of the lesson/other*
12. Have you ever experienced physical discomfort or strain from piano playing? If yes, please specify. *Yes, often/yes occasionally/yes, rarely/no, never*
13. Have you ever experienced psychological problems related to piano playing? If yes, please specify. *Yes, often/yes occasionally/yes, rarely/no, never*
14. Have you ever used any supporting programs, apps, or other digital tools for your piano lessons or at home practice? If yes, please specify. *Yes, often/yes occasionally/yes, rarely/no, never*
15. If you have used supporting technologies (question above), how helpful would you rate them? *Very/somewhat/barely/not at all*
16. If you didn't find them useful, why not? *(Free text form)*
17. In general, would you describe yourself as open to new technology? *Very open/open/ moderately open/hardly open/not at all open*
18. How would you rate your learning pace? (students only) *Very slow/slow/average/fast/ very fast*
19. Are you satisfied with your learning pace? (students only) *Very satisfied/satisfied/average/ less satisfied/not satisfied at all*

20. Have you ever noticed further problems or difficulties during piano lessons not mentioned here before? *(Free text form)*
21. Did your piano lessons continue during the COVID-19 lockdown? *Yes/no/limited*
22. If you had lessons during the COVID-19 lockdown, what form did they take? (Presence/other, which communication tools were used?) *(Free text form)*
23. How satisfied were you with your lessons during the lockdown? *Very satisfied/satisfied/neutral/less satisfied/not at all satisfied*
24. What problems or difficulties did you encounter in your piano lessons during that time? *(Free text form)*

**Appendix B**

*Final Code Structure*
Scenarios
Distance learning
Visualization of musical output
Performance analysis
Body movement analysis
Technology acceptance
Pragmatic aspects
    Functionality
    Usability
Hedonic aspects
Evaluation
Positive impression (sentiment)
Negative impression (sentiment)
High priority
Low priority
Role and level of expertise
Student
    Beginner
    Intermediate/Expert
Teacher
    Beginner
    Intermediate/Expert
Status quo
Teacher–student interaction
Self-directed learning
Pandemic-related online lessons
Motivation and needs
Physical and psychological stress
Learning and teaching problems
Technology usage
Piano playing skills
    External focus
    Teaching methods and procedure
    Playing by the notes
    Body posture and movements
    Musical imagination
    Articulation and tone quality
    Rhythm
    Music theory
    Aural training

## Appendix C

**Table A1.** Participants' responses to questionnaire. Responses are presented as a percentage on questions relating to physical and psychological complaints as well as experiences with digital support systems and feedback preferences related to piano playing. Multiple answer choices were possible.

| | Preference (%) | |
|---|---|---|
| | **Teachers** | **Students** |
| **1. What form of practice do your students (for teachers) or you (for students) prefer?** | | |
| Exploratory free practice | 9 | 50 |
| Clear instructions and quick feedback | 73 | 50 |
| Other | 18 | 0 |
| **2. What type of feedback do you think is useful in piano lessons?** | | |
| Interrupt and point out errors directly when they occur | 27 | 38 |
| Feedback directly after the respective exercise piece | 55 | 63 |
| Feedback only at the end of the lesson | 0 | 0 |
| Other | 18 | 13 |
| **3. Have you already experienced physical stress or overstrain due to playing the piano? [1]** | | |
| Yes, often | 27 | 13 |
| Yes, occasionally | 45 | 25 |
| Yes, rarely | 18 | 50 |
| No, never | 9 | 13 |
| **4. Have you ever suffered from mental or emotional discomfort while playing the piano? [2]** | | |
| Yes, often | 9 | 38 |
| Yes, occasionally | 54 | 25 |
| Yes, rarely | 9 | 0 |
| No, never | 27 | 38 |
| **5. Have you (for students) or your students (for teachers) used any additional supporting programs/apps/other digital learning tools for the classroom? [3]** | | |
| Yes, often | 9 | 13 |
| Yes, occasionally | 27 | 0 |
| Yes, rarely | 0 | 13 |
| No, never | 64 | 75 |
| **6. Would you generally describe yourself as open to new technologies?** | | |
| Yes, very | 36 | 50 |
| Yes | 9 | 0 |
| Average | 36 | 25 |
| Less | 18 | 25 |
| Not at all | 0 | 0 |

[1] If yes, what was the nature of the problem (back pain; slight pain in wrist, neck, upper arms/shoulder, and lumbago; tendon sheath inflammation; pain in the upper arm; tendinitis). [2] If yes, what was the nature of the problem (stress; anxiety; stage fright). [3] Which apps/programs (Medly; Piano Notes Pro; Piano games; MyEarTraining/Perfect Ear; metronome; Skype; WhatsApp; Zoom; video recordings).

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
