# Peer review of "What Early User Involvement Could Look Like—Developing Technology Applications for Piano Teaching and Learning"

_mti, doi:10.3390/mti5070038_

Round 1
Reviewer 1 Report
The authors' review of the article is satisfactory from my perspective.
The authors have better articulated some parts of the article, making it more fluent to read. The English has been revised and corrected, as well as some parts of the introduction and state of the art, including more contributions.
Considering my first revision and the work done by the authors for this second version, I recommend the article for publication.
Reviewer 2 Report
All my comments have been considered and addressed by the authors.
Author Response
Please see attached document for details.

This manuscript is a resubmission of an earlier submission. The following is a list of the peer review reports and author responses from that submission.
Round 1
Reviewer 1 Report
This paper presents the results of an online survey with 19 pianists and interviews with 5 pianists about three hypothetical scenarios for the use of 'Tactile Internet with Human in the Loop (TaHIL)' technologies to promote piano learning. The responses seem to be a mixed bag, with pianists identifying certain challenges of teaching and learning and expressing enthusiasm for some ideas while showing practical and experiential reservations about others. The authors discuss these responses in the context of whether TaHIL technologies can assist piano learning.
The domain of technology-assisted instrumental learning is an interesting and active one, and this work is a first step towards making a contribution in that area. However, as it stands, the proposed scenarios are too vague, and the discussion too speculative, for the results to be of immediate use to other researchers. Below I discuss some of my specific concerns. Overall I believe this paper will need to go further in actually implementing and evaluating at least one specific scenario before it will be ready for publication.
The paper opens with a discussion of TaHIL technologies, which proposes a more human-centred approach to the Tactile Internet. TaHIL is defined mainly in generalities -- "wearable devices with the fast and reliable transmission of multi-modal information"; "devices that provide real-time feedback on playing performance or hand and body movement"; "smart sensors"; "intelligent communication networks"; machine learning; visual and haptic feedback. The paper argues for the novelty of TaHIL, but to the extent it's possible to understand the specifics, all of these ideas have been around for many years. MIDI, movement sensing (using various modalities), audio analysis, visual and haptic feedback, score following, timing and rhythm analysis have all been used for technologically-assisted instrumental learning for a long time. Other than aggregating existing technologies under one umbrella, it's not clear what is new about TaHIL in this context.
The paper then discusses a human-centred approach to technology design. This is a welcome focus, indeed a necessity in this domain. However the paper sets up a straw man (ll. 57-58): "This is not usually done in basic research, where often technological innovations are led by technical requirements and constraints." The authors should be more clear what kind of "basic research" this statement refers to, as nearly every application of technology to instrumental performance and pedagogy will have considered human factors to at least a certain degree, and evaluation with performers is also standard. Beyond music, human-computer interaction research has explored usability, and in the past 10-20 years has moved beyond task-based metrics toward a more nuanced view of technology incorporated into everyday life (third-wave HCI), considering its experiential, social and cultural aspects (some of the hedonic elements described in the paper). Research through design (RtD) is one subcomponent of this reframing, with knowledge generated through the design process itself. That literature can be an excellent source of support for the authors' arguments, and a few such references do appear e.g. [56], but the arguments especially in Section 1.1 are too focused on a critique that does not accurately describe current research in the area.
After some discussion about the potential benefits of technology for piano learning (Section 1.2), the paper sets out three scenarios which will be presented to participants in the surveys and interviews. I appreciate that the nature of the paper is to discuss ideas with pianists rather than to implement any specific technology, and so the fact that details of implementation are not mentioned here is not by itself a problem.
However, the scenarios still have a couple drawbacks. First, they are not particularly novel at a conceptual level. Visual feedback of body kinematics (Scenario I) has been fairly extensively explored in instrumental learning (perhaps more for violin than piano). Visualisation of note-level performance (Scenario II) has been tried a great many times since at least the 1980's, in forms including keyboards with light-up keys, piano-roll style visualisations, Guitar Hero-type games indicate what keys to press, etc. Scenario II might go beyond this: "audio data of several performance parameters is analysed in (quasi) real-time using machine learning, providing information on how the sound was produced and how it can be improved." (ll. 225-226), but this description is overly vague even as a prompt for discussion.
Second, the scenarios sometimes contradict what is known about instrumental learning. For example, reference [45] cited in Scenario III to justify directing attention to an external focus (which is a good idea) also suggests that instant visual feedback is typically distracting and unhelpful for performance, which would seem to undermine Scenario I. There are also issues of building dependency on feedback (the training wheels effect) that should be considered.
The procedure for the surveys and interviews are sound and described in a good level of detail. With 19 participants, the quantitative analysis is perhaps less interesting than the qualitative findings from the 5 interviews. Some of the comments are intriguing, but my sense is that many comments will be familiar, even obvious, to pianists and piano researchers (e.g. importance of relaxation; individuality of performance; independence of hands; use of metronomes). I also wonder about the usefulness of some of the responses, e.g. ll. 508-509 describing the piano-roll visualisation idea: "the idea that there are notes shown and that you then press the key on the piano and that feedback comes back: was it the right key or not? I think that's very cool."
This ultimately points to a larger issue, which is that the responses of a relatively small number of pianists to hypothetical technology prompts is unlikely to yield much insight in comparison to what we already know from the existing body of literature which evaluates specific technologies with musicians. Speaking from personal experience, I often find that musicians are enthusiastic about technology in the abstract, whether out of the novelty effect or social factors with regard to the experimenter, but that in practice, the problematic aspects or limitations of technologies only become manifest when they are used in practice, often over an extended time.
This is perhaps why there are so many proposed technological solutions to instrument learning that never left the research lab, or which were discontinued from the market after a short time -- not that human factors were never considered, but rather that getting it right is extremely difficult, dependent on dozens of subtle details both technological and social. Moving this field forward needs a more specific framing, either implementing a specific technology, mocking it up in a robust way (e.g. 'Wizard of Oz' style design fiction), or at least digging beyond first impressions in a longer-term speculative design exercise. As it stands, I see the current paper as a pilot study for such a project, not complete in its current form but perhaps suggesting directions that could be developed further.
Reviewer 2 Report
Overall the article is well-written if rather overlong and it is, at times -- particularly in the Introduction -- excessively detailed. The sample size is rather small and the research is of a rather speculative nature. Having said that, the results are of interest and the thematic analyses have been well-performed and the findings expand knowledge.
I recommend some minor editing to reduce the length of Sections 1 and 1.1. -- just keep what is essential to the story.
Reference should be made to Scenario IV somewhere in the Introduction -- perhaps directly after Scenario III -- because it comes as a surprise when first mentioned in the Results section.
Overall the quality of the writing was good but there are a number of minor issues with phrasing that I, as a first English language speaker, can easily spot. I detail all of these below.
l.50: COVIS-19 -> COVID-19
1.51: lilitations -> limitations
l.51 - l.54: “Further examples are devices that provide real-time feedback on playing performance or hand and body movement corrections to prevent playing-related injuries [4]. This can be highly relevant for democratising access to music skills and experiences, i.e., anyone can have the opportunity to learn music playing regardless of age, sex, and geographical location [5].” This “democratising” will only occur if the device is sufficiently cheap and is part of a network that is sufficiently free-to-access. If anything, such technology seems more likely to increase global inequalities rather than reduce them. So, I think this should be phrased more cautiously.
l.77: “will allow to integrate” -> “will allow the integration of”
l.105: “mostly a self-fulfilment” is not grammatically correct in this context.
l.134-135: “Especially designer’s skills to communicate abstract ideas through visualizations or 134 prototypes facilitate the exchange of researchers with diverse stakeholders involved [17].” Clumsy English — rephrase.
l.148: “societies” -> “society’s”
l.156: “This user’s 156 feedback” which user is being referred to here?
l.183: “For example, machine learning can provide an objective rating of the playing quality”. I suspect that “automatic” is a more appropriate word here than “objective”
l.205: “how TaHIL applications for piano may look like” “what .. looks like” is the correct English not “how .. looks like” (this is a common error made by non-first-language-English writers)
l.239: “Further, it could support pianists on stage, where keeping an external focus is especially crucial to avoid black-outs.” The term black-outs should be explained on first use— the term commonly means unconsciousness but that is clearly not what is meant here!
l.259: when you say “issues” do you mean “problems”? (I realise that issues typically mean problems in a coding context but in the context of a paper as broad as this the meaning is ambiguous).
l.312: “94:21,2” ?
l.325: “Status Quo” -> “status quo”
l.354-364: this should be in Sec. 2.1, not here.
l.383: wrong indentation
l.441: “Pianist’s” -> “Pianists’”
l.472: “blackouts” — that word again — what exactly is meant — forgetting?
l.609: “artistical” -> “artistic”
l.737: “complimentary” -> “complementary”
l.739: “acceptance in” -> “acceptance by”
Reviewer 3 Report
I commend the authors of this manuscript for the research approach taken, particularly, in actively involving the users (piano teachers and students) in the process of considering and reviewing proposed approaches for development of TaHIL applications in piano pedagogy contexts.
Here are a few minor points which you may wish to consider:
- As a highly experienced piano teacher myself, across all skill levels (teaching for more than 25 years), I anticipate that TaHIL tools' level of meaningfulness will also relate to the age of learners, skill level and specific intended outcomes that they are aimed for, from a perspective of what solution/s they can offer to particular problems. And it seems to me that all the piano students involved were all adults and not exactly at a beginner stage - it might be worth specifying this in the abstract.
- In page 2, line 50, replace COVIS with COVID.
- I understand that Scenario IV came about as a result from participants' inputs. However, as it definition only appears later, the reader is not sure of what it refers to in the results section (Figure 4). I wonder if this scenario could be described and graphically represented either: a) after the description given of scenario III in page 6? Perhaps side-by-side to a reference that it was developed through participants' ideas, etc? or b) In a brief way, where it appears in the results presentation - so that readers can understand what that is, side-by-side to the results presentation.
- In the Discussion Section, a reference in brackets to the specific scenario side-by-side to the title of each sub-section, would be helpful, from a readers' perspective . E.g. Analysis and Processing of Body Movement Through Wearable Sensors (scenario I).
- I have not seen considerations on the limitations of the study. Would what I mentioned here in 1 be a limitation? What other limitations could there be? (e.g. small sample of participants; all from the same geographical context and musical tradition?)
Reviewer 4 Report
The paper presents design insights from both piano teachers and learners about the potential application of tactile multimodal technologies to the field of instrument learning. The topic presented and the scenarios designed are interesting and I think coherent with the purpose of the Journal. The paper is generally well-written; the methodologies as well as weaknesses of the study well highlighted but English is sometimes misspelled, thus I suggest a general language revision. The standard deviation of the population reached for the study is generally very high, so I recommend increasing the number of subjects in the study, if possible.